# Experimental and Numerical Study on Rapid Evacuation Characteristics of Staircases in Campus Buildings

**Qian Zhang [1,2], Fei Yu [3], Shan Gao [4,\*], Chen Chang [3] and Xusheng Zhang [5]**

1    Xi'an Changda Asset Management Co., Ltd., Chang'an University, Xi'an 710064, China; zq563178082@163.com
2    The Engineering Design Academy of Chang'an University Co., Ltd., Xi'an, Xi'an 710064, China
3    School of Civil Engineering, Xijing University, Xi'an 710123, China; yufeidickyyufei@163.com (F.Y.); cc10135mm@163.com (C.C.)
4    Harbin Institute of Technology, School of Civil Engineering, Harbin 150090, China
5    China Jikan Research Institute of Engineering Investigations and Design Co., Ltd., Xi'an 710043, China; zhangxs1423@163.com
\*    Correspondence: gaoshan@hit.edu.cn

**Abstract:** In this work, we conducted downward evacuation experiments in four types of staircases under various smoke visibility conditions of the naked eye, wearing sunglass and wearing eyeshades. Ten male and ten female college students were recruited to conduct the evacuation as a single male, single female, two males supporting one another, two females supporting one another and one male carrying another on his back. The evacuation time on each floor was recorded. The corresponding evacuation models were established by Pathfinder and verified against the test data. The effects of evacuation crowd density and response time considering gender factors on the evacuation time were simulated using the models. The results show that under the experimental condition of low visibility, the curve of evacuation time presents a stable state, whose change with the increase in the floors is not obvious. The increase in the evacuation time under different visibility indicates that males have better adaptability to the environment than females. The curves of SSP (straight running stairs with platform) and DSS (double split parallel stairs) are smoother than those of DPS (double running parallel stairs) and CS (corner stairs), indicating less pressure and less congestion during evacuation. During the emergency evacuation, the crowd pressure on the platform of the staircases is small. The front section of the flight and the corner part of the staircases are prone to congestion during evacuation. Under the influence of gender factors, since the response time of males is longer than that of females, the smaller the proportion of males, the smaller the time growth rate considering the reaction time. With the increase in crowd density, the effect of response time on total evacuation time becomes smaller.

**Keywords:** staircase; evacuation; gender; visibility; campus building

## 1. Introduction

Emergency safety evacuation is a necessary public protection measure to reduce the consequences of major accidents. In case of a fire, earthquake or other dangerous situations in buildings, stairs are the most important channel in emergency safety evacuation [1,2]. Considerable earthquake damage data show that the evacuation capacity of staircase directly affects the life safety of individuals [3,4]. Campus buildings at universities are the main places for students to live and study, and for teachers and staff to work. In case of danger, a large number of teachers and students flow into the stairwell, resulting in large instantaneous flow and congestion, which reduces the evacuation efficiency and causes secondary injuries such as stampede events. In most countries, fire law stipulates that elevators cannot be used during fire evacuation; therefore, evacuation through staircases is more important.

Numerous studies have been conducted on the evacuation of people from buildings. In 1986, the fire evacuation drill conducted by Kagawa et al. [5] showed that simultaneous total evacuation was impractical and that more frequent education on fire safety was necessary. Through a full-scale test series, Jensen [6] and Proulx et al. [7] concluded that the visibility distance is an important factor in conditions with extreme smoke. Shields et al. [8] conducted an interview with 9/11 evacuees which indicated that more attention needs to be given to fire safety staff training programs. Xu and Song [9] developed an improved multi-grid model for staircase evacuation, where the rectangular body size, various walking speeds in different densities and turning behavior of pedestrians are taken into account. Recently, Agyemang and Kinateder [10] summarized that biomechanical analyses of pedestrian staircase descent add nuance by characterizing factors relevant to safe movement on stairs, such as foot placement, the use of handrails and balance. In China, relevant studies on the evacuation simulation in staircases have also been conducted in the past decade. Yuan et al. [11] concluded that the staircase position affected the evacuation performance in a university dormitory. Li et al. [12] compared the evacuation performance in four types of staircases by simulation and concluded that the single-run and no-platform staircases are prone to congestion at the upper and lower ends of the flight. Li et al. [13] developed a causation model of accidents and the analysis results showed that safety awareness, safety cognition and fear had a great influence on unsafe behaviors. The evacuation simulation conducted by Wang et al. [14] implied that too many handrails in the main staircase will increase the overall evacuation time. Wang and Wen [15] carried out an evacuation simulation and suggested that stairway evacuation capacity is not affected by the width of stairs.

It can be seen from the literature review that the experimental research data on evacuation through staircases in university campus buildings are relatively limited and cannot fully reflect the evacuation characteristics of staircases in campus buildings. The results of various evacuation simulations are inconsistent to some extent, such as the effect of handrails [10,14] and staircase types [12,15]. Moreover, the physical and safety awareness profiles of evacuees are normally not considered in the simulations and tests. These problems bring some difficulties to the formulation of evacuation plans in universities. In addition, to cooperate with the interconnection between building groups, more and more styles of staircases are used. However, few studies, especially experimental studies, focus on the comparative analysis regarding the evacuation capacity of different staircases.

In this work, using four kinds of common staircases in campus buildings, evacuation tests were carried out, considering the effects of gender, smoke visibility and evacuation object. Based on the simulation analysis validated against the test results, we studied the characteristics of different staircases to improve the efficiency of emergency safety evacuation and provide a reference for the design of campus buildings and emergency plans.

## 2. Types of Staircases in Campus Buildings

At a private university in northwest China, the four most common types of staircases in teaching buildings on the campus were selected to conduct the evacuation experiments. These are straight running stairs with platforms, double running parallel stairs, corner stairs and double split parallel stairs, as shown in Figure 1.

(a). Straight running stairs with platforms (SSP): an intermediate rest platform is arranged to connect the upper and lower floors without changing the direction. The staircase has a simple structure, but occupies a large amount of linear space. It is suitable for outdoor stairs of teaching buildings and stadiums.

(b). Double running parallel stairs (DPS): two flights are usually of equal length to save area and occupy less linear space. It is the most widely used type. When going up and down multi-story floors, it can save traffic areas and shorten the walking distance compared to straight running stairs. It is suitable for dormitory buildings, office buildings and cafeterias.

(c). Corner staircase (CS): it occupies less space and can be used in the corner of the building. The direction of people flow can be changed through the platform. It is suitable for teaching buildings, cafeterias and libraries.

(d). Double split parallel staircase (DSS): this kind of staircase is evolved based on double running parallel stairs. When the width of the stairs is large, it is usually divided into two sections. It is suitable for libraries, teaching buildings and office buildings.

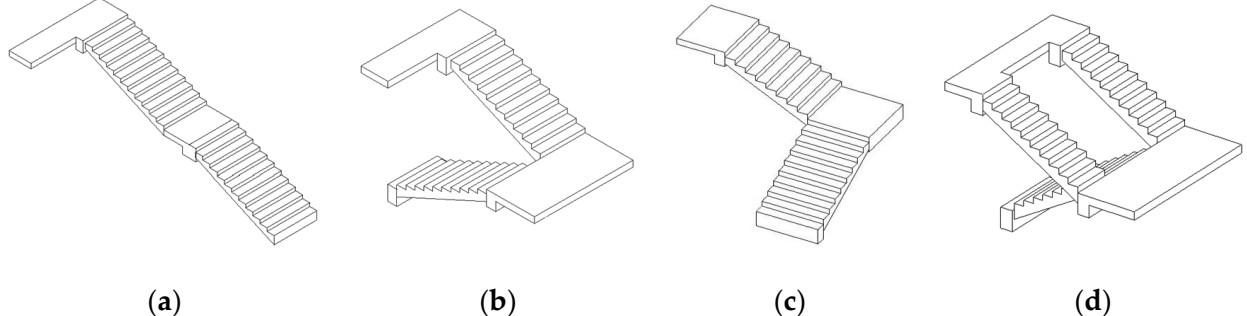

<div align="center">(<b>a</b>)          (<b>b</b>)          (<b>c</b>)          (<b>d</b>)</div>

**Figure 1.** Four types of common staircases. (**a**) Straight running stairs with platform (SSP); (**b**) double running parallel stairs (DPS); (**c**) corner stairs (CS); (**d**) double split parallel stairs (DSS).

## 3. Evacuation Test of Staircase in Campus Building

### 3.1. Evacuation Scenario

Figure 2 presents the basic information of four staircases [16,17] and evacuation direction. Straight running stairs with platform, spiral double running parallel stairs and L-shape corner stairs include six floors, and the U-shape double split parallel staircase is five floors. The height of the ground floor is 3.9 m and that of the other floors is 3.6 m. Each flight includes 12 risers, except for the ground floor, which has 13 risers. The rise height and depth of each riser are 0.15 m and 0.3 m, respectively, resulting in a 27-degree slope, as shown in Figure 3. The width of the riser is consistent with that of the flight, as shown in Figure 2.

The objects of the evacuation experiment are a single male (SM), single female (SF), two males supporting one another (TM), two females supporting one another (TF) and one male carrying another on his back (MC). The data of individuals' evacuation via the specific staircase were collected to analyze the micro-mechanism of people's movement in an emergency evacuation. In addition, smoke visibility will affect the evacuation rate of individuals in the staircase. Variables regarding smoke visibility set in the experiment are listed in Table 1. Medium and low visibility levels were simulated by the participants wearing sunglass and eyeshades. Figure 4 shows the evacuation test site of the staircase.

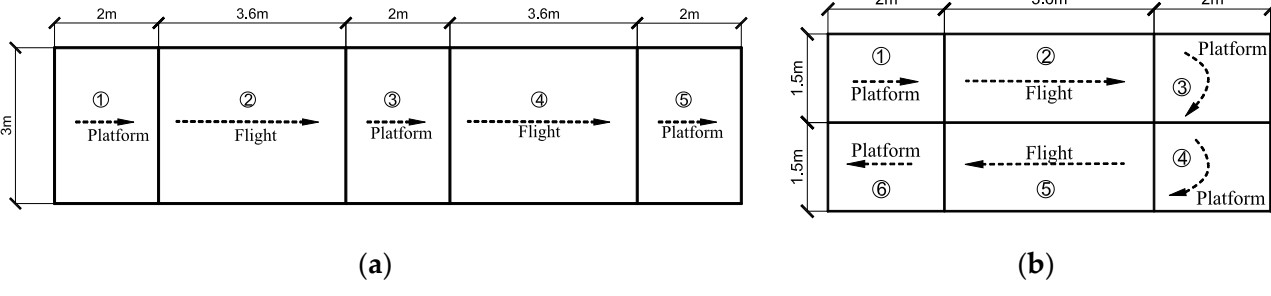

<div align="center">(<b>a</b>)                                                (<b>b</b>)</div>

**Figure 2.** *Cont*.

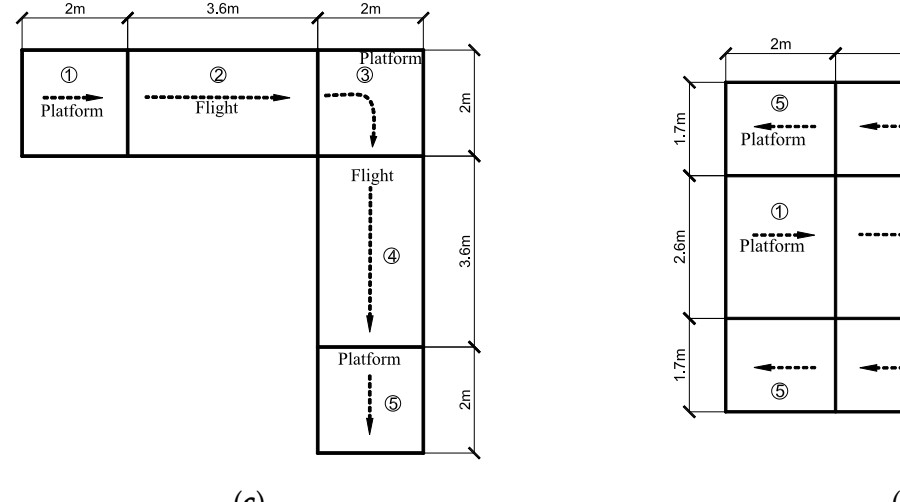

(**c**)                           (**d**)

**Figure 2.** Basic information of the staircases. (**a**) Straight running stairs with platform (SSP), six floors; (**b**) double running parallel stairs (DPS), six floors; (**c**) corner stairs (CS), six floors; (**d**) double split parallel stairs (DSS), five floors.

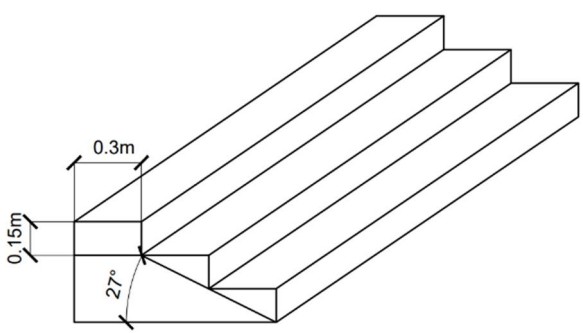

**Figure 3.** Height and depth of the riser.

**Table 1.** Test scenarios.

| Smoke Visibility | High | Medium | Low |
|---|---|---|---|
| Simulated scenario | Naked eye | Wearing sunglass | Wearing eyeshade |

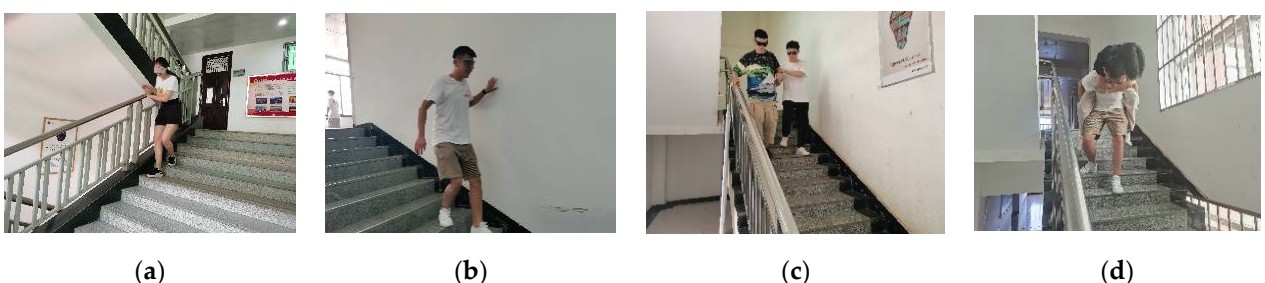

(**a**)           (**b**)           (**c**)           (**d**)

**Figure 4.** Photos at the test scene. (**a**) SF under low visibility; (**b**) SM under low visibility; (**c**) TM under medium visibility; (**d**) MC under medium visibility.

The evacuation experiments for four types of staircases were carried out under the same conditions. To obtain the evacuation time in the staircase only, during the experiment, all participants were evacuated directly from the top platform of the staircase, rather than from a classroom from the top floor to the ground floor. In other words, the evacuation behavior of the participants in the classroom and the hallway was not considered in the test.

After obtaining approval from the institutional review board, the participants were selected and signed a consent form for the evacuation test. At first, some volunteers were recruited randomly by using an online mobile application of the campus population, which was 10,000. Then, 10 male and 10 female student participants were selected for the experiment based on the gender and physical profile of the students. Each participant was numbered during the preparation of the experiment. A camera was placed on each floor of the evacuation experiment to record the behavior of participants during the evacuation. Each experiment was monitored by a timekeeper to record the evacuation time.

### 3.2. Test Results

Table 2 shows the total evacuation time of the five types of test objects in four types of staircases under three degrees of visibility. The values in the table represent the range of the evacuation time of all the participants. Figure 5 presents the detailed evacuation times on each floor. The evacuation time of the females was slightly longer than that of the males. With the decrease in visibility, all experimental participants needed more time to evacuate, and the range of the evacuation time gradually expanded. The increase in the evacuation time of the female students under different visibility was larger than that of the male students, which indicates that males have better adaptability to the environment than females, and that females are more affected by the visibility on the stairs than males.

**Table 2.** Total evacuation time.

| Object Scenario | Time/s | SM (Single Male) | SF (Single Female) | TM (Two Males) | TF (Two Females) | MC (One Male Carrying) |
|---|---|---|---|---|---|---|
| SSP | High | 30.5–35.5 | 32.8–44.1 | 33.7–40.0 | 36.0–44.2 | 40.3–51.5 |
| | Medium | 34.3–39.4 | 38.2–45.6 | 36.0–42.4 | 40.0–48.2 | 42.4–54.8 |
| | Low | 55.0–62.5 | 62.1–75.3 | 59.9–70.4 | 63.2–72.1 | 75.7–98.1 |
| DPS | High | 34.8–40.1 | 42.9–49.8 | 42.1–50.1 | 44.8–56.5 | 49.2–61.0 |
| | Medium | 37.2–45.3 | 42.6–50.7 | 48.2–54.4 | 50.2–59.3 | 49.0–62.4 |
| | Low | 57.8–63.5 | 62.3–71.48 | 66.3–77.5 | 75.5–87.1 | 94.6–108.7 |
| CS | High | 46.1–51.0 | 48.9–55.2 | 48.7–53.0 | 50.9–58.3 | 60.1–64.3 |
| | Medium | 45.4–52.1 | 52.4–60.0 | 52.0–57.6 | 56.9–64.3 | 63.0–70.1 |
| | Low | 98.9–115.9 | 127.7–147.2 | 108.3–121.9 | 140.2–156.3 | 157.8–178.3 |
| DSS | High | 27.0–30.9 | 32.0–37.0 | 31.3–39.4 | 34.2–43.3 | 35.3–44.5 |
| | Medium | 28.4–31.1 | 32.0–38.9 | 34.2–40.2 | 35.9–44.2 | 39.0–47.8 |
| | Low | 44.6–52.6 | 51.8–60.0 | 50.0–59.3 | 52.9–63.4 | 65.2–77.3 |

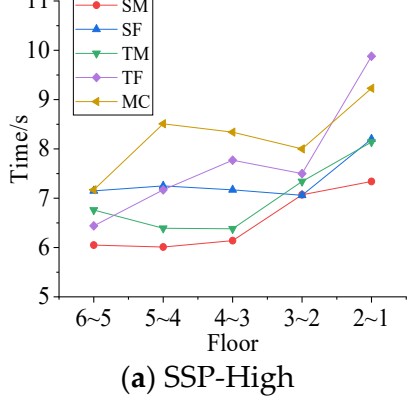

(**a**) SSP-High

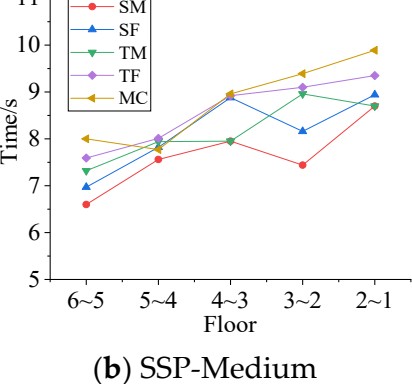

(**b**) SSP-Medium

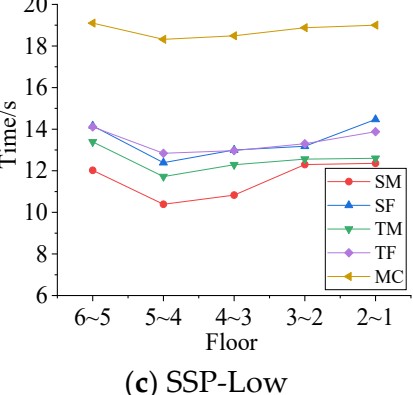

(**c**) SSP-Low

**Figure 5.** *Cont*.

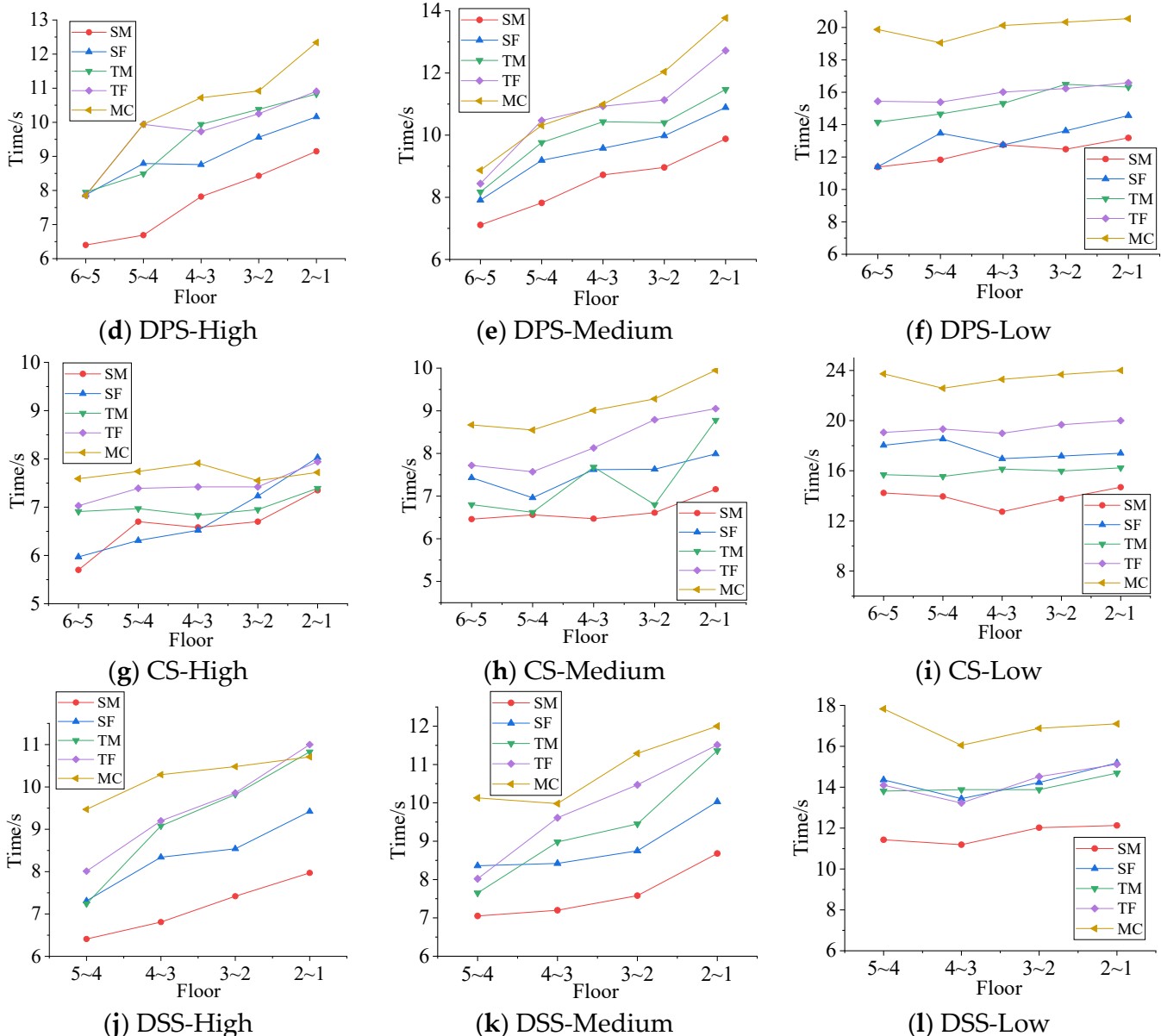

**Figure 5.** Time distribution of various staircases and scenarios.

Under the normal conditions and wearing sunglasses, the average evacuation time shows an upward trend with the increase in the floors, while under the experimental conditions of wearing eyeshades, the curve of the evacuation time on each floor shows a stable state whose change with the increase in descending floors is not very obvious. Additionally, according to the observation in the test site, the utilization rate of handrails is significantly increased under the experimental conditions of wearing sunglasses and eye masks.

We may conclude that, under normal conditions, with the increase in downward evacuation time, the physical ability of the participants increases. However, under the condition of wearing eyeshades, the decrease in the visibility slows the speed of the participants, resulting in the decrease in their physical ability. Therefore, the corresponding curves show a stable state. It can be seen from Figure 5 that the evacuation time becomes longer when the participants go down the initial floor under the condition of wearing eyeshades, which may result from the fact that the participants are not familiar with the stairs in complete darkness at the beginning of the evacuation process.

## 4. Development and Validation of Simulation Model

### 4.1. Model Development

To verify whether the experimental data and simulation data of the evacuation time are consistent, we used Pathfinder to establish the staircase model diagram of four stair types, as shown in Figure 6. Since the object of this section is the influence of staircase type on evacuation time, only the staircase parts in the campus building were developed without considering other factors, and the flights are connected by platforms. All the dimensions of the staircases in the simulation are identical to the measured data in the experiment. "Steering mode" from Pathfinder, which is the most commonly used mode in evacuation simulations, was chosen to present the law of movement characteristics in the test.

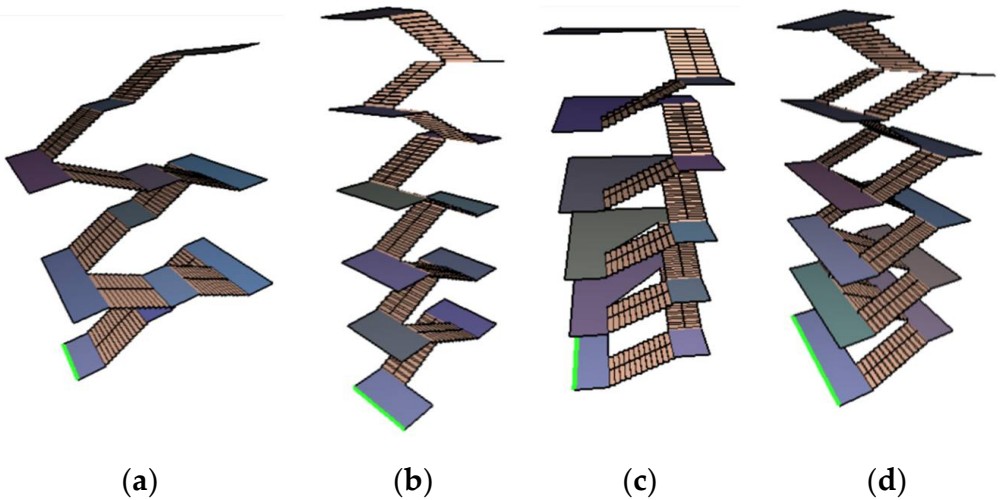

(**a**)          (**b**)          (**c**)          (**d**)

**Figure 6.** Simulation model of four staircases. (**a**) Straight running stairs with platform (SSP); (**b**) double running parallel stairs (DPS); (**c**) corner stairs (CS); (**d**) double split parallel stairs (DSS).

### 4.2. Model Validation

The individual evacuation rates were collected by timekeepers according to the stair parameters and statistical evacuation time in the experiment. After modeling the staircases, the individual evacuation rates collected from the test were input into the Pathfinder software. The individual speed is the average speed of each participant category in the real data of the experiment. The participant parameters were set according to GB/T 10000-1988 [18], and the data of males and females aged 18–55 in the 50th percentile were selected. The average height of males and females was 167.8 cm and 157.0 cm, respectively, and the shoulder width of males and females was 37.5 cm and 35.1 cm, respectively. Each simulation takes the average results of five runs, which avoids the influence of randomness on the simulation results. Table 3 shows the comparison between the median values of the tested total evacuation time and simulated total evacuation time, which shows a good agreement.

**Table 3.** Comparison of tested and simulated data.

| Scenarios | Object / Time/s | Data Type | SM (Single Male) | SF (Single Female) | TM (Two Males) | TF (Two Females) | MC (One Male Carrying) |
|---|---|---|---|---|---|---|---|
| SSP | High | Test | 32.6 | 36.8 | 35.0 | 38.7 | 41.2 |
| | | Simulation | 32.3 | 36.7 | 35.1 | 38.2 | 42.0 |
| | Medium | Test | 38.2 | 40.7 | 40.8 | 42.9 | 44.0 |
| | | Simulation | 37.8 | 40.5 | 41.2 | 42.7 | 44.1 |
| | Low | Test | 57.9 | 67.2 | 62.5 | 67.1 | 93.7 |
| | | Simulation | 57.5 | 62.6 | 62.7 | 67.4 | 93.2 |
| DPS | High | Test | 38.4 | 45.1 | 47.5 | 48.6 | 51.7 |
| | | Simulation | 38.1 | 45.6 | 47.1 | 48.5 | 51.3 |
| | Medium | Test | 42.4 | 47.5 | 50.2 | 53.6 | 55.9 |
| | | Simulation | 42.3 | 47.9 | 50.9 | 53.5 | 55.7 |
| | Low | Test | 61.6 | 65.8 | 76.9 | 79.6 | 99.9 |
| | | Simulation | 62.3 | 65.3 | 76.4 | 79.5 | 99.2 |
| CS | High | Test | 48.6 | 50.1 | 50.1 | 52.2 | 61.2 |
| | | Simulation | 48.7 | 50.9 | 50.6 | 52.6 | 61.7 |
| | Medium | Test | 50.2 | 57.6 | 55.6 | 58.2 | 65.4 |
| | | Simulation | 50.9 | 57.0 | 55.5 | 58.1 | 65.5 |
| | Low | Test | 104.9 | 138.4 | 111.3 | 145.0 | 167.2 |
| | | Simulation | 104.3 | 138.7 | 111.4 | 145.5 | 166.9 |
| DSS | High | Test | 28.6 | 33.6 | 36.9 | 38.0 | 40.9 |
| | | Simulation | 29.3 | 33.4 | 36.2 | 38.7 | 40.1 |
| | Medium | Test | 30.5 | 35.5 | 37.4 | 39.6 | 43.4 |
| | | Simulation | 30.1 | 36.0 | 37.3 | 39.2 | 43.6 |
| | Low | Test | 46.7 | 56.9 | 56.2 | 57.2 | 67.8 |
| | | Simulation | 46.3 | 56.3 | 56.7 | 57.3 | 67.3 |

## 5. Parametric Analysis

### 5.1. Crowd Density

To analyze the effect of crowd density on the evacuation time, 10 to 100 evacuees at the interval of 10 people were adopted in the simulation. The average evacuation speed from the tested data was 1.7 m/s, obtained from the ratio between the total distance in the stairwell and the total evacuation time.

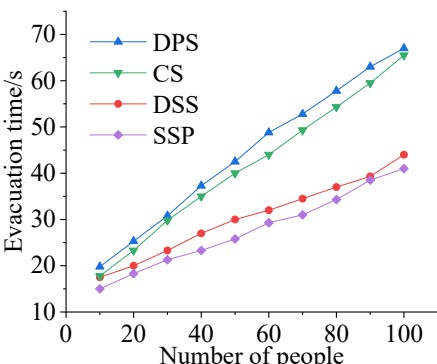

**Figure 7.** Effect of crowd density.

As shown in Figure 7, the evacuation time of the four stairs increases linearly with the increase in the total number of evacuees. In general, the longest evacuation time is from DPS (double running parallel stairs), and the shortest evacuation time is from SSP (straight running stairs with platform). When the number of evacuees is 10, the evacuation times of SSP (straight running stairs with platform), DPS (double running parallel stairs), CS (corner stairs) and DSS (double split parallel stairs) are 15 s, 19.8 s, 17.8 s and 17.5 s, respectively. The maximum difference between the evacuation times is 4.8 s and the minimum difference is 2.5 s. When the number of evacuees is 50, the evacuation times of SSP, DPS, CS and DSS are 25.8 s, 42.5 s, 40 s and 30 s, respectively, and the maximum and minimum differences

are 16.7 s and 2.5 s, respectively. When the number of evacuees is 100, the evacuation times of SSP, DPS, CS and DSS are 41 s, 67 s, 65.5 s and 44 s, respectively, and the maximum and minimum differences are 26 s and 1.5 s, respectively. We concluded that evacuation through straight running stairs with platforms and double split parallel stairs is faster than through double running parallel stairs and corner stairs.

### 5.2. Real-Time Crowd Flow

The simulated real-time crowd flow curves of single-layer downward evacuation of 50 evacuees are shown in Figure 8. The curves of SSP and DSS are relatively smooth, which indicates less pressure and less congestion during the evacuation. The flow curves of DPS and CS fluctuate, indicating that congestion occurs from time to time.

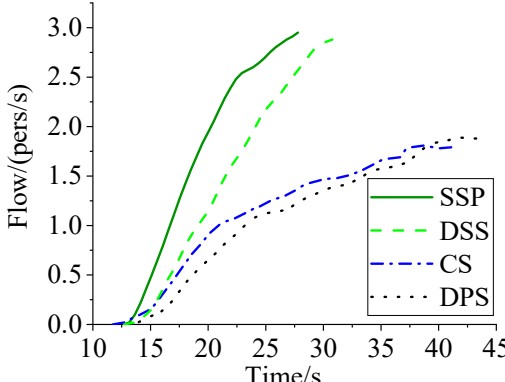

**Figure 8.** Real-time crowd flow of 50 evacuees on one floor.

The crowd density distribution of 50 and 100 evacuees in the four staircases is shown in Figure 9. It can be seen that the locations prone to congestion accidents vary according to different types of staircases. The starting point of each flight in SSP is prone to congestion, while there is no congestion at the platform position. The corners of DPS and DSS are prone to congestion, but thanks to the characteristics of diversion, the second flight of DSS is far less prone to congestion than DPS. In other words, compared with DPS, the probability of dangerous accidents of DSS is relatively low. The upper end of the first flight in CS is prone to congestion, but the congestion condition of its platform is acceptable.

### 5.3. Response Time

Individual safety awareness affects the response time before evacuation, which is also a key factor affecting the total evacuation time. However, most simulation models in previous studies do not consider the psychological effects of individual safety awareness on evacuation behavior.

Referring to the response time of emergency evacuation obtained in Ref. [19] through experiments, and the previous study from the authors' group [20] regarding the gender factors in college evacuation, which found that male students have lower scores on safety behavior than female students, the response times of males and females in the simulation were set as 5 s and 2 s, respectively. According to the experimental average value, the evacuation speeds of males and females were set as 2.0 m/s and 1.5 m/s, respectively. Other participant parameters are identical to those in the last section. The crowd density of 50, 100 and 150 evacuees was simulated. The effect of response time on evacuation time is shown in Figure 10. Since the initial speed of the males is greater than that of the females, the greater the proportion of males, the shorter the evacuation time. However, the response time of the males is longer than that of the females, so the smaller the proportion of the males, the smaller the time growth rate considering the response time. Moreover, with the increase in crowd density, the effect of response time on the total evacuation time becomes smaller.

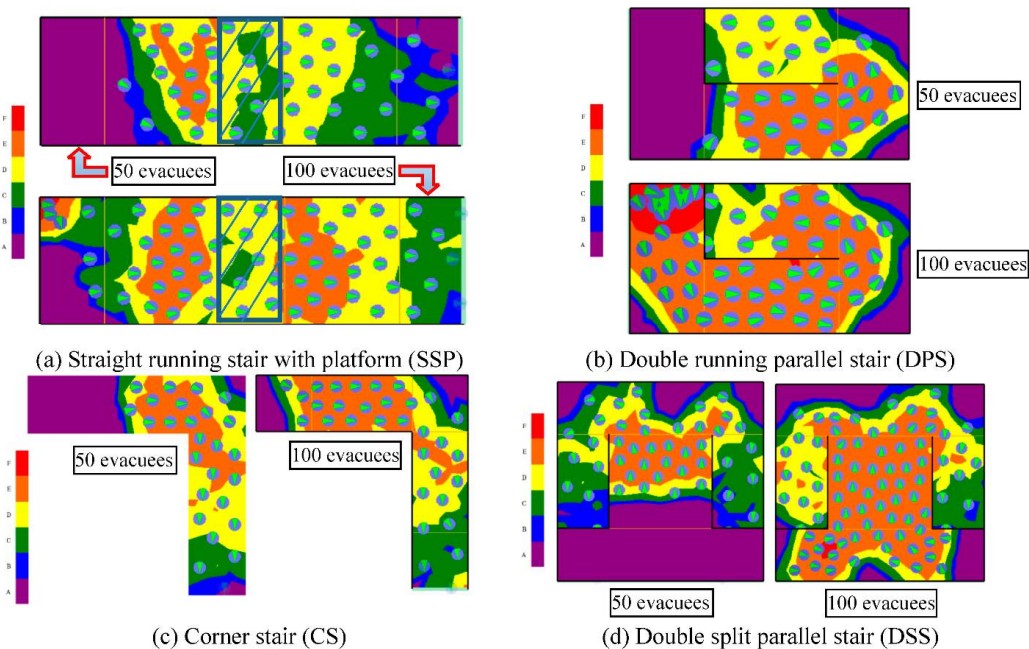

**Figure 9.** Crowd density distribution in staircases.

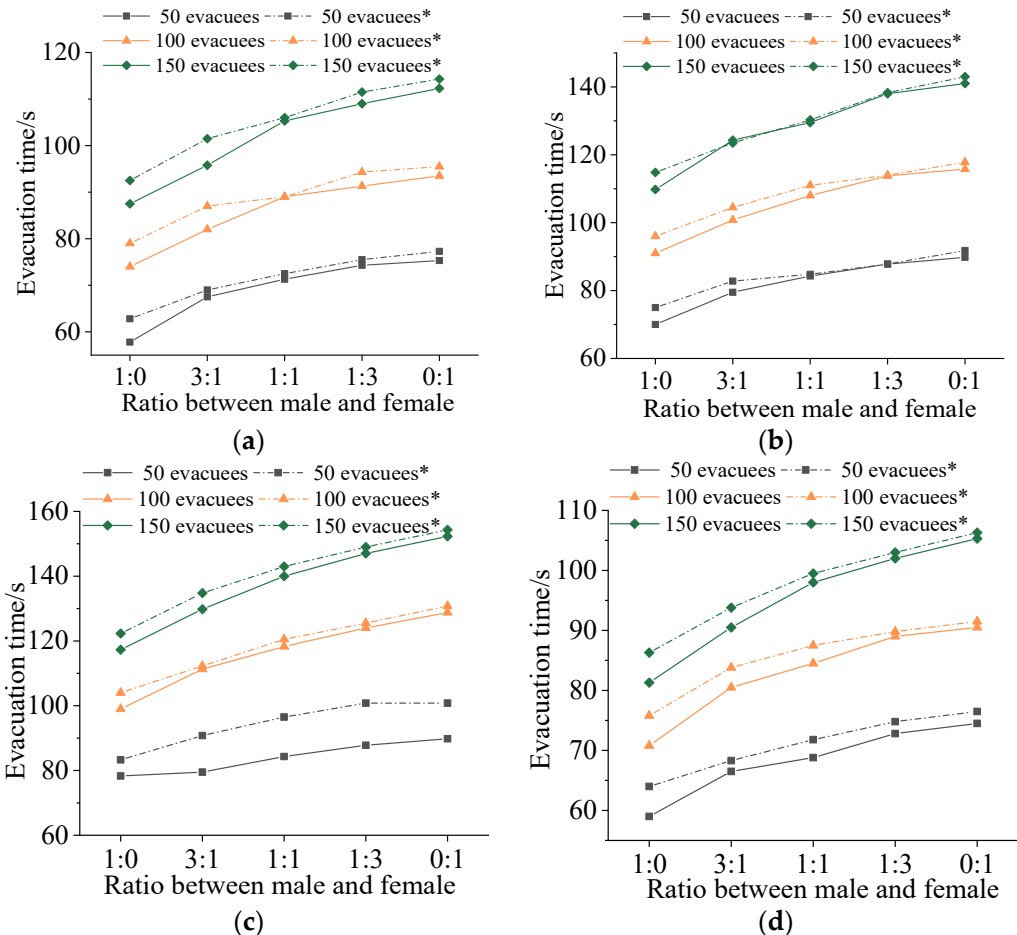

**Figure 10.** Effect of response time. (**a**) Straight running stairs with platform (SSP); (**b**) double running parallel stairs (DPS); (**c**) corner stairs (CS); (**d**) double split parallel stairs (DSS). Note: "*" means the simulation considering the response time.

## 6. Conclusions

In this study, we carried out downward evacuation experiments in four types of staircases under various visibility conditions. Compared to the previous studies, the effects of gender and evacuation object were considered in the tests. The corresponding evacuation models were also established and verified. The effects of evacuation crowd density and especially the response time considering gender factors on the downward evacuation time in four kinds of staircases were simulated. Our main conclusions are as follows.

(1) In the process of single-person downward evacuation, the evacuation time of females is longer than that of males, and the evacuation time range of females is also wider than that of males. Under the experimental condition of low visibility (wearing eyeshades), the curve of evacuation time presents a stable state, whose change with the increase in the floors is not obvious.

(2) The increase in the evacuation time under different visibility conditions indicates that males have better adaptability to the environment than females. Under the condition of medium and low visibility, the probability of using handrails in the evacuation is significantly higher, indicating that the handrails are necessary during emergency evacuation, especially in public buildings.

(3) The simulation curves of SSP (straight running stairs with platform) and DSS (double split parallel stairs) are smoother than those of DPS (double running parallel stairs) and CS (corner stairs), indicating less pressure and less congestion during evacuation in the first two types of staircases. During emergency evacuation simulation, the crowd pressure on the platform of the staircases was small. The front section of the flight and the corner part of the staircases are prone to congestion during evacuation, confirming the positive effect of platforms on evacuation.

(4) Under the influence of gender factors, since the response time of males is longer than that of females, it can be seen from the curve that the smaller the proportion of males, the smaller the time growth rate considering the reaction time. With the increase in crowd density, the effect of response time on total evacuation time becomes smaller.

Considering the limitations of this study, such as small sample size, age group and crowding degree, more tests are still needed in future studies. Specifically speaking, since only students are considered in this study, the effect of age group is not studied, and college professors and researchers account for a portion of the population in campus buildings whose action capability may be weaker than that of students. Additionally, the crowding degree is not considered in this study, since it is dangerous to conduct a crowd evacuation test in a staircase. These data may only be obtained in a real accident evacuation. Thus, it is necessary to install some monitoring facilities at important exits. Moreover, the predicted evacuation time should also be compared with the required evacuation time (such as fire spread time and structure failure time) [21] according to different types of building structures.

**Author Contributions:** Methodology, Q.Z.; Funding acquisition, Q.Z.; Writing—original draft, F.Y., C.C. and S.G.; Conceptualization, F.Y. and S.G.; Investigation, C.C.; Visualization, Resources, X.Z. All authors have read and agreed to the published version of the manuscript.

**Funding:** The project was supported by the National Natural Science Foundation of China (no. 51908085), the Natural Science Foundation of Chongqing (cstc2020jcyj-msxmX0010), Fundamental Research Funds for the Central Universities (AUGA5710010322 and 2020CDJ-LHZZ-013) and the Youth Innovation Team of Shaanxi Universities (21JP138), which are gratefully acknowledged.

**Informed Consent Statement:** Informed consent was obtained from all subjects involved in the study.

**Data Availability Statement:** The raw/processed data required to reproduce these findings cannot be shared at this time as the data also form part of an ongoing study.

**Acknowledgments:** The authors would like to appreciate the help from the participants.

**Conflicts of Interest:** We declare that we do not have any commercial or associative interest that represents a conflict of interest in connection with the work submitted.

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
