# Peer review of "Experimental and Numerical Study on Rapid Evacuation Characteristics of Staircases in Campus Buildings"

_buildings, doi:10.3390/buildings12060848_

Round 1

Reviewer 1 Report

Dear Authors

This valuable paper is well written, but it is better to address the comments below. 

Abstract

The effects of evacuation crowd density and response time considering gender factors on the evacuation time are simulated.

After the author’s sentence, it is important to inform the reader on what is the methodology/methods used in simple straightforward way on how data modelled/analysed. As most of the abstract has become a result presentation only, rather than listing the paper structure.

Introduction

The results show that the overall evacuation efficiency is affected by different stair forms, stair width, stair spacing, handrails and the number of evacuees.

Clear and obvious but requires a ref.

Type of staircase in campus buildings

A camera was placed on each floor of the evacuation experiment to record the behavior of people during evacuation. Each experiment was followed by a timekeeper to record the time.

From the beginning to this sentence, it hasn’t been mentioned how many FLOORS the paper has covered. It is scientifically robust to let the academic researchers know the number of floors avoiding research repletion.

1.1. Test results

Table 2. Total evacuation time

Was the total evacuation time for all participants (20 males and females) per experiment?

4.1. Model development

Table 3. Comparison of tested and simulated data.

Were the results in Table 3 total time of evacuation?

5.1. Crowd density

The average evacuation

speed from the tested data is used as 1.7 m/s.

Reference required with literature preferences.

kindest regards, 

Reviewer.

Reviewer 2 Report

What a great topic?! I congratulate the authors for addressing this critical issue. However, I have the following suggestions/ directions/ feedback.

·        Under various “visibilities”, what do authors mean by visibilities (maybe slightly elaborate).

·        The abstract jumps to results quickly without mentioning the method. What type of simulations were run? Software name? How many times? How many conditions were introduced?

·         I am not sure about other parts of the world but in the US, we use different terminologies for staircase types like “L-shape, U-shape, straight, spiral..”. In this paper, it was slightly difficult to visualize the staircases with these new terminologies. Are these common terminologies? When we say “corner staircase”, we refer to staircases that are in the corner of the building regardless of their shapes.

·         In Introduction: campus buildings are not just for teachers and students. Staff and visitors should be included.

·        The use of elevators during a fire is prohibited in most countries (of course some elevators have exceptions).

·        In the second paragraph, what do the authors mean by “visibility” level? Are they referring to the visibility of the staircase or the overall sense of visibility?

·         When the authors say that the results were inconsistent, it would be helpful to cite and reference two examples of these inconsistencies. Also, I don’t know whether the authors brought something new by examining different types of staircases, has this been included in other studies. This will be a good opportunity for the authors to reference here.

·        I had difficulty understanding section 3.1. (need to be proofread) but I believe I got the point of the author. I don’t know what the authors mean by the word ladder in the paragraph (spelled Lader in the figure). Do they mean to say “flight”?! Ladder in our world means a vertical structure with steps. So the authors need to spend time here and linguistically/ grammatically make the necessary revision. Also, The depth and width (.15m/.3m) didn’t make sense to me. From looking at the same numbers in the figure, maybe the authors mean to say the rise height (.15m) and depth (.3).

·         How did the authors obtain personnel movement (law of movement characteristics).

·         Up until page 3, I was under the assumption that the authors ran “computer or virtual reality” simulations to simulate fire evacuations. I think that the authors of the abstract should explicitly say that it is a real-life simulation and should mention also in the abstract how many participants were recruited. They then need to elaborate on how the participants were selected (on page 3). Currently, the authors say that the participants in the experiment were randomly selected from 10 male and 10 female students from the campus. It is impossible that the population of the campus is just 20. I think that the authors recruited 10 from (whatever the male student population is). The authors need to give us some demographics about the campus population (undergraduate and graduates) etc.

·         Also, only in the last paragraph of page 3, I begin to understand what the authors meant by “visibility”. I think they wanted to simulate “smoke” visibility and not the visibility of exit signs and staircases.

·         Also, now that I understand that this is a real simulation, I need to know where these staircases are located and where did the participants begin the experiment. Were they inside the classroom? Were they all equally distanced?

·         When I arrived to page 5, I got further confused. I believe that the authors recruited 10 participants to collect data on behavior. Is this true? I believe that they are trying to either use it for the simulation model or to compare it with the simulation model.

·         In several places, the authors used the word “boy”. What do the authors mean by “boy”. In the US, boy is “a male child from birth to adulthood”. I am confused because the experiment was on male students at a university campus.

·         I could not understand the results, so I am not going to comment on the results. I didn’t understand the results because 1) the study is missing the literature review part, 2) did not describe the data collection methods well, 3) as per my previous comment, I wasn’t sure how the real simulation was used in the model.  The research methods are not clearly articulated.

·         There are limitations to the study that the authors need to address such as small sample size, age group, etc

Round 2

Reviewer 2 Report

Page 1. The staircase is relatively closed. In the US firecodes allow for open external staircases and I am trying to understand what it means to say "closed". It is very important that the authors speak about fire codes in their setting. How many exits are required by codes and what are the features that regulate these stairs/exits.  Also, elsewhere in the methods where they ran the experiment, what are the codes applied to their setting and whether these were equally implemented in the model. We have learnt for example from simulations of exits at the tragic 911 that they were not enough but in their simulation, the input was precise and captured both the number of occupants and the number of coded exits. 

3.1. It is called the "riser".

What do authors mean in Fig 2 by adding "six-floor" to the caption

Annotate figure 3 (width, riser)

The authors still need to elaborate on the recruitment process. Was a flyer sent out to the participants or were they friends of the author. Did you go through the IRB process/ consent form?! Were they explained the objectives?

Overall, the research is missing key background review for the authors to be able to connect their results to during the discussion. I am not going to comment on the discussion, it is fine but it doesn't give the reader any room to judge whether this research contributed anything new. There are numerous simulations on fire evaculations, how do the authors stand out in how they designed their research and in the findings. This is a traditional requirement across all research papers (especially quantitative studies like this one).

The limitation paragraph need to be expanded beyond one sentence. When they say more tests are needed, what do they really mean and why. Usually the limitation section will help guide future research and with a short statement, it is difficult to understand these challenges.

In the results and conclusions, there was no discussion around the findings. Why male students had better adaptability to the environment. Anything in the research help the authors support their findings? Again, this goes back to the missing background literature. Without it, this study will be classified as a report that typically list their findings as opposed to a research where findings are supported by literature to explain why they found what they found.

I think the authors need at least two to three weeks to address this gap and look back at their work to make it stronger. Again, I enjoyed reading about their experiments, such a good contribution but I assure you that the readers will not find it useful if there are no explanation and they will not be able to judge if it is a new contribution.

Author Response

Please see the response in the attachment

Round 3

Reviewer 2 Report

I thank the authors for this much improved manuscript. I have now very few comments (mostly around the conclusion). The rest of the comments can be easily done if the authors have access to an editor to make the paragraphs flow and read better.

Here they are:

I still think that “the staircase is relatively enclosed..” is still unclear and confusing. Maybe the authors can get away with deleting it unless they want to explain how relevant the statement is to the study.

In the second paragraph, maybe add “Numerous studies or numerous researchers have been conducted…”

Also, the addition of the new second paragraph is excellent, maybe in the stage of editing, some transitions could be added to enhance the flow of reading. It is an excellent addition but it is a lot of information. Again, this could be enhanced during editing.

The authors mention: “from the perspective of safety, the experiment in this paper focuses on single person..” but they focus also on one carrying another person, so it is not a single person unless they mean something else. To avoid confusion, just remove this statement.

Instead of saying “after going through IRB process”, the authors should be clear that they “obtained approval from IRB..”. Both are different but important. And also add that participants were signed a consent form to participate.

During editing, more attention should be given to make the section 3.2 read and flow better. The same goes for section 5.3

It feels that the conclusion section is more of a discussion of results section. Also, once the discussion section is separately added, the authors need to connect what they found to literature review. Then the conclusions could be more specific to the contributions of the study, the summary of findings, the importance of staircases and handrails and ending with limitations.
